# A Simple and Effective Baseline for Out-of-Distribution Detection using an Abstention Class

## Abstract

Refraining from confidently predicting when faced with categories of inputs different from those seen during training is an important requirement for the safe deployment of deep learning systems. While simple to state, this has been a particularly challenging problem in deep learning, where models often end up making overconfident predictions in such situations. In this work we present a simple, but highly effective approach to deal with out-of-distribution detection that uses the principle of abstention: when encountering a sample from an unseen class, the desired behavior is to abstain from predicting. Our approach uses a network with an extra abstention class and is trained on a dataset that is augmented with an uncurated set that consists of a large number of out-of-distribution (OoD) samples that are assigned the label of the abstention class; the model is then trained to learn an effective discriminator between in and out-of-distribution samples. We compare this relatively simple approach against a wide variety of more complex methods that have been proposed both for out-of-distribution detection as well as uncertainty modeling in deep learning, and empirically demonstrate its effectiveness on a wide variety of of benchmarks and deep architectures for image recognition and text classification, often outperforming existing approaches by significant margins. Given the simplicity and effectiveness of this method, we propose that this approach be used as a new additional baseline for future work in this domain.

## 1 Introduction and Related Work

Most of supervised machine learning has been developed with the assumption that the distribution of classes seen at train and test time are the same. However, the real-world is unpredictable and open-ended, and making machine learning systems robust to the presence of unknown categories and out-of-distribution samples has become increasingly essential for their safe deployment. While refraining from predicting when uncertain should be intuitively obvious to humans, the peculiarities of DNNs makes them overconfident to unknown inputs Nguyen et al. (2015) and makes this a challenging problem to solve in deep learning.

A very active sub-field of deep learning, known as *out-of-distribution* (OoD) detection, has emerged in recent years that attempts to impart to deep neural networks the quality of "knowing when it doesn't know". The most straight-forward approach in this regard is based on using the DNNs output as a proxy for predictive confidence. For example, a simple baseline for detecting OoD samples using thresholded softmax scores was presented in Hendrycks & Gimpel (2016). where the authors provided empirical evidence that for DNN classifiers, in-distribution predictions do tend to have higher winning scores than OoD samples, thus empirically justifying the use of softmax thresholding as a useful baseline. However this approach is vulnerable to the pathologies discussed in Nguyen et al. (2015). Subsequently, increasingly sophisticated methods have been developed to attack the OoD problem. Liang et al. (2018) introduced a detection technique that involves perturbing the inputs in the direction of increasing the confidence of the network's predictions on a given input, based on the observation that the magnitude of gradients on in-distribution data tend to be larger than for OoD data. The method proposed in Lee et al. (2018) also involves input perturbation, but confidence in this case was measured by the Mahalanobis distance score using the computed mean and covariance of the pre-softmax scores. A drawback of such methods, however, is that it introduces a number of

hyperparameters that need to be tuned on the OoD dataset, which is infeasible in many real-world scenarios as one does not often know in advance the properties of unknown classes. A modified version of the perturbation approach was recently proposed in in Hsu et al. (2020) that circumvents some of these issues, though one still needs to ascertain an ideal perturbation magnitude, which might not generalize from one OoD set to the other.

Given that one might expect a classifier to be more uncertain when faced with OoD data, many methods developed for estimating uncertainty for DNN predictions have also been used for OoD detection. A useful baseline in this regard is the temperature scaling method of Guo et al. (2017) that was was proposed for calibrating DNN predictions on in-distribution data and has been observed to also serve as a useful OoD detector in some scenarios. Further, label smoothing techniques like *mixup* Zhang et al. (2017) have also been shown to be able to improve OoD detection performance in DNNs Thulasidasan et al. (2019). An ensemble-of-deep models approach, that is also augmented with adversarial examples during training, described in Lakshminarayanan et al. (2017) was also shown to improve predictive uncertainty and succesfully applied to OoD detection.

In the Bayesian realm, methods such as Maddox et al. (2019) and Osawa et al. (2019) have also been used for OoD detection, though at increased computational cost. However, it has been argued that for OoD detection, Bayesian priors on the data are not completely justified since one does not have access to the prior of the open-set Boult et al. (2019). Nevertheless, simple approaches like dropout – which have been shown to be equivalent to deep gaussian processes Gal & Ghahramani (2016) – have been used as baselines for OoD detection.

Training the model to recognize unknown classes by using data from categories that do not overlap with classes of interest has been shown to be quite effective for out-of-distribution detection and a slew of methods that use additional data for discriminating between ID and OD data have been proposed. DeVries & Taylor (2018) describes a method ithat uses a separate confidence branch and misclassified training data samples that serve as a proxy for OoD samples. In the outlier exposure technique described in Hendrycks et al. (2018), the predictions on natural outlier images used in training are regularized against the uniform distribution to encourage high-entropy posteriors on outlier samples. An approach that uses an extra-class for outlier samples is described in Neal et al. (2018), where instead of natural outliers, *counterfactual* images that lie just outside the class boundaries of known classes are generated using a GAN and assigned the extra class label. A similar approach using generative samples for the extra class, but using a conditional Variational Auto-Encoders Kingma & Welling (2013) for generation, is described in Vernekar et al. (2019). A method to force a DNN to produce high-entropy (i.e., low confidence) predictions and suppress the magnitude of feature activations for OoD samples was discussed in Dhamija et al. (2018), where, arguing that methods that use an extra background class for OoD samples force all such samples to lie in one region of the feature space, the work also forces separation by suppressing the activation magnitudes of samples from unknown classes

The above works have shown that the use of *known* OoD samples (or *known unknowns*) often generalizes well to *unknown unknown* samples. Ineed, even though the space of unknown classes is potentially infinite, and one can never know in advance the myriad of inputs that can occur during test time, empirically this approach has been shown to work. The abstention method that we describe in the next section borrows ideas from many of the above methods: as in Hendrycks et al. (2018), we uses additional samples of real images and text from non-overlapping categories to train the model to abstain, but instead of entropy regularization over OoD samples, out method *uses an extra abstention class*. While it has been sometimes argued in the literature that that using an additional abstention (or rejection) class is not an effective approach for OoD detection Dhamija et al. (2018); Lee et al. (2017), comprehensive experiments we conduct in this work demonstrate that this is not the case. Indeed, we find that such an approach is not only simple but also highly effective for OoD detection, often outperforming existing methods that are more complicated and involve tuning of multiple hyperparameters. The main contributions of this work are as follows:

- To the best of our knowledge, this is the first work to comprehensively demonstrate the efficacy of using an extra abstention (or rejection class) in combination with outlier training data for effective OoD detection.

- In addition to being effective, our method is also simple: we introduce no additional hyperparameters in the loss function, and train with regular cross entropy. From a practical standpoint, this is especially useful for deep learning practitioners who might not wish

to make modifications to the loss function while training deep models. In addition, since outlier data is simply an additional training class, no architectural modifications to existing networks are needed.

- Due to the simplicity and effectiveness of this method, we argue that this approach be considered a strong baseline for comparing new methods in the field of OoD detection.

## 2 OUT-OF-DISTRIBUTION DETECTION WITH AN ABSTAINING CLASSIFIER (DAC)

Our approach uses a DNN trained with an extra *abstention* class for detecting out-of-distribution and novel samples; from here on, we will refer to this as the deep abstaining classifier (DAC). We augment our training set of in-distribution samples ($\mathcal{D}_{in}$) with an auxiliary dataset of *known* out-of-distribution samples ($\tilde{\mathcal{D}}_{out}$), that are known to be mostly disjoint from the main training set (we will use $\mathcal{D}_{out}$ to denote *unknown* out-of-distribution samples that we use for testing). We assign the training label of $K+1$ to all the outlier samples in $\tilde{\mathcal{D}}_{out}$ (where $K$ is the number of known classes) and train with cross-entropy; the minimization problem then becomes:

$$\min_{\theta} \mathbb{E}_{(\mathbf{x},y)\sim\mathcal{D}_{in}} \left[-\log P_\theta(y=\hat{y}|\mathbf{x})\right] + \mathbb{E}_{\mathbf{x},\mathbf{y}\sim\tilde{\mathcal{D}}_{out}} \left[-\log P_\theta(y=K+1|\mathbf{x})\right] \tag{1}$$

where $\theta$ are the weights of the neural network. This is somewhat similar to the approaches described in Hendrycks et al. (2018) as well as in Lee et al. (2017), with the main difference being that in those methods, an extra class is not used; instead predictions on outliers are regularized against the uniform distribution. Further the loss on the outlier samples is weighted by a hyperparameter $\lambda$ which has to be tuned; in contrast, our approach does not introduce any additional hyperparameters.

In our experiments, we find that the presence of an abstention class that is used to capture the mass in $\tilde{\mathcal{D}}_{out}$ significantly increases the ability to detect $\mathcal{D}_{out}$ during testing. For example, in Figure 1, we show the distribution of the winning logits (pre-softmax activations) in a regular DNN (left). For the same experimental setup, the *abstention logit* of the DAC produces near-perfect separation of the in and out-of-distribution logits indicating that using an abstention class for mapping outliers can be a

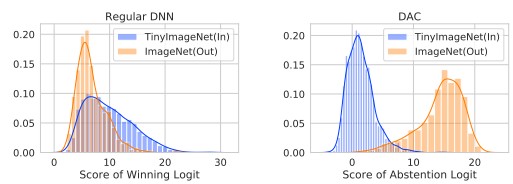

Figure 1: An illustration of the separability of scores on in and out-of-distribution data for a regular DNN (left) and the DAC (right).

very effective approach to OoD detection. Theoretically, it might be argued that the abstention class might only capture data that is aligned with the weight vector of that class, and thus this approach might fail to detect the myriad of OoD inputs that might span the entire input region. Comprehensive experiments over a wide variety of benchmarks described in the subsequent section, however, empirically demonstrate that while the detection is not perfect, it performs very well, and indeed, much better than more complicated approaches.

Once the model is trained, we use a simple thresholding mechanism for detection. Concretely, the detector, $g(x): X \to 0,1$ assigns label 1 (OoD) if the softmax score of the abstention class, i.e., $p_{K+1}(x)$ is above some threshold $\delta$, and label 0, otherwise:

$$g(\mathbf{x}) = \begin{cases} 1 & \text{if } p_{K+1}(x) \geq \delta \\ 0 & \text{otherwise} \end{cases} \tag{2}$$

Like in other methods, the threshold $\delta$ has to be determined based on acceptable risk that might be specific to the application. However, using performance metrics like area under the ROC curve (AUROC), we can determine threshold-independent performance of various methods, and we use this as one of our evaluation metrics in all our experiments.

## 3 EXPERIMENTS

The experiments we describe here can be divided into two sets: in the first set, we compare against methods that are explicitly designed for OoD detection, while in the second category, we compare

against methods that are known to improve predictive uncertainty in deep learning. In both cases, we report results over a variety of architectures to demonstrate the efficacy of our method.

## 3.1 DATASETS

For all computer vision experiments, we use CIFAR-10 and CIFAR-100 Krizhevsky & Hinton (2009) as the in-distribution datasets, in addition to augmenting our training set with 100K unlabeled samples from the Tiny Images dataset Torralba et al. (2008). For the out-of-distribution datasets, we test on the following:

- **SVHN** Netzer et al. (2011), a large set of $32 \times 32$ color images of house numbers, comprising of ten classes of digits $0 - 9$. We use a subset of the 26K images in the test set.
- **LSUN** Yu et al. (2015), the Large-scale Scene Understanding dataset, comprising of 10 different types of scenes.
- **Places365** Zhou et al. (2017), a large collection of pictures of scenes that fall into one of 365 classes.
- **Tiny ImageNet** tin (2017) (not to be confused with Tiny Images) which consists of images belonging to 200 categories that are a subset of ImageNet categories. The images are $64 \times 64$ color, which we scale down to $32 \times 32$ when testing.
- **Gaussian** A synthetically generated dataset consisting of $32 \times 32$ random Gaussian noise images, where each pixel is sampled from an i.i.d Gaussian distribution.

For the NLP experiments, we use 20 Newsgroup Lang (1995), TREC Sherman, and SST Socher et al. (2013) datasets as our in-distribution datasets, which are the same as those used by Hendrycks et al. (2018) to facilitate direct comparison. We use the 50-category version of TREC, and for SST, we use binarized labels where neutral samples are removed. For out OoD training data, we use unlabeled samples from Wikitext2 by assigning them to the abstention class. We test our model on the following OoD datasets:

- **SNLI** Bowman et al. (2015) is a dataset of predicates and hypotheses for natural language inference. We use the hypotheses for testing .
- **IMDB** Maas et al. (2011) is a sentiment classification dataset of movie reviews, with similar statistics to those of SST.
- **Multi30K** Barrault et al. (2018) is a dataset of English-German image descriptions, of which we use the English descriptions.
- **WMT16** Bojar et al. (2016) is a dataset of English-German language pairs designed for machine translation task. We use the English portion of the test set from WMT16.
- **Yelp** Zhang et al. (2015) is a dataset of restaurant reviews.

## 3.2 COMPARISON AGAINST OOD METHODS

In this section, we compare against a slew of recent state-of-the-art methods that have been explicitly designed for OoD detection. For the image experiments, we compare against the following:

- **Deep Outlier Exposure**, as described in Hendrycks et al. (2018) and discussed in Section 1
- **Ensemble of Leave-out Classifiers** Vyas et al. (2018) where each classifier is trained by leaving out a random subset of training data (which is treated as OoD data), and the rest is treated as ID data.
- **ODIN**, as described in Liang et al. (2018) and discussed in Section 1. ODIN uses input perturbation and temperature scaling to differentiate between ID and OoD samples.
- **Deep Mahalanobis Detector**, proposed in Lee et al. (2018) which estimates the class-conditional distribution over hidden layer features of a deep model using Gaussian discriminant analysis and a Mahalanobis distance based confidence-score for thresholding, and further, similar to ODIN, uses input perturbation while testing.

- **OpenMax**, as described in Bendale & Boult (2016) for novel category detection. This method uses mean activation vectors of ID classes observed during training followed by Weibull fitting to determine if a given sample is novel or out-of-distribution.

For all of the above methods, we use published results when available, keeping the architecture and datasets the same as in the experiments described in the respective papers. For the NLP experiments, we only compare against the published results in Hendrycks et al. (2018). For OpenMax, we re-implement the authors' published algorithm using the PyTorch framework Paszke et al. (2019).

### 3.2.1 METRICS

Following established practices in the literature, we use the following metrics to measure detection performance of our method:

- **AUROC** or Area Under the Receiver Operating Characteristic curve depicts the relationship between the True Positive Rate (TPR) (also known as Recall)and the False Positive Rate (FPR) and can be interpreted as the probability that a positive example is assigned a higher detection score than a negative example Fawcett (2006). Unlike 0/1 accuracy, the AUROC has the desirable property that it is not affected by class imbalance[1].
- **FPR at 95% TPR** which is the probability that a negative sample is misclassified as a positive sample when the TPR (or recall) on the positive samples is 95%.

In work that we compare against, the out-of-distribution samples are treated as the positive class, so we do the same here, and treat the in-distribution samples as the negative class.

### 3.2.2 RESULTS

Detailed results against the various OoD methods are shown in Tables 1 through 3 for vision and language respectively, where we have a clear trend: in almost all cases, the DAC outperforms the other methods, often by significant margins especially when the in-distribution data is more complex, as is the case with CIFAR-100. While the Outlier Exposure method Hendrycks et al. (2018) (shown at the top in Table 1) is conceptually similar to ours, the presence of an extra abstention class in our model often bestows significant performance advantages. Further, we do not need to tune a separate hyperparameter which determines the weight of the outlier loss, as done in Hendrycks et al. (2018).

In fact, the simplicity of our method is one of its striking features: we do not introduce any additional hyperparameters in our approach, which makes it significantly easier to implement than methods such as ODIN and the Mahalanobis detector; these methods need to be tuned separately on each OoD dataset, which is usually not possible as one does not have access to the distribution of unseen classes in advance. Indeed, when performance of these methods is tested without tuning on the OoD test set, the DAC significantly outperforms methods such as the Mahalanobis detector (shown at the bottom of Table 1). We also show the performance against the OpenMax approach of Bendale & Boult (2016) in Table 2 and in every case, the DAC outperforms OpenMax by significant margins.

While the abstention approach uses an extra class and OoD samples while training, and thus does incur some training overhead, it is significantly less expensive during test time, as the forward pass is no different from that of a regular DNN. In contrast, methods like ODIN and the Mahalanobis detector require gradient calculation with respect to the input in order to apply the input perturbation; the DAC approach thus offers a computationally simpler alternative. Also, even though the DAC approach introduces additional network parameters in the final linear layers (due to the presence of an extra abstention class), and thus might be more prone to overfitting, we find that this to be not the case as evidenced by the generalization of OoD performance to different types of test datasets.

### 3.3 COMPARISON AGAINST UNCERTAINTY-BASED METHODS

Next we perform experiments to compare the OoD detection performance of the DAC against various methods that have been proposed for improving predictive uncertainty in deep learning. In these cases,

---

[1]An alternate area-under-the-curve metric, known as Area under Precision Recall Curve, or AUPRC, is used when the size of the negative class is high compared to the positive class. We do not report AUPRC here, as we keep our in-distribution and out-of-distribution sets balanced in these experiments.

vs. Outlier Exposure (OE) Hendrycks et al. (2018)
(Model: Wide ResNet 40x2)

| $\mathcal{D}_{out}$ | $\mathcal{D}_{in}$: CIFAR-10 | | | | $\mathcal{D}_{in}$: CIFAR-100 | | | |
|---|---|---|---|---|---|---|---|---|
| | FPR95 $\downarrow$ | | AUROC $\uparrow$ | | FPR95 $\downarrow$ | | AUROC $\uparrow$ | |
| | OE | Ours | OE | Ours | OE | Ours | OE | Ours |
| SVHN | 4.8 | $\mathbf{2.0}_{0.69}$ | 98.4 | $\mathbf{99.46}_{0.16}$ | $\mathbf{42.9}$ | $40.46_{11.96}$ | $\mathbf{86.9}$ | $85.44_{6.25}$ |
| LSUN | 12.1 | $\mathbf{0.1}_{0.06}$ | 97.6 | $\mathbf{99.96}_{0.02}$ | 57.5 | $\mathbf{9.27}_{4.62}$ | 83.4 | $\mathbf{97.67}_{1.40}$ |
| Places365 | 17.3 | $\mathbf{0.22}_{0.12}$ | 96.2 | $\mathbf{99.92}_{0.05}$ | 49.8 | $\mathbf{23.37}_{5.30}$ | 86.5 | $\mathbf{93.98}_{1.67}$ |
| Gaussian | 0.7 | $\mathbf{0.13}_{0.14}$ | 99.6 | $\mathbf{99.93}_{0.09}$ | $\mathbf{12.1}$ | $13.26_{14.74}$ | 95.7 | $90.03_{11.81}$ |

vs. Ensemble of Leave-out Classifiers (ELOC) Vyas et al. (2018)
(Model: Wide ResNet 28x10)

| $\mathcal{D}_{out}$ | $\mathcal{D}_{in}$: CIFAR-10 | | | | $\mathcal{D}_{in}$: CIFAR-100 | | | |
|---|---|---|---|---|---|---|---|---|
| | FPR95 $\downarrow$ | | AUROC $\uparrow$ | | FPR95 $\downarrow$ | | AUROC $\uparrow$ | |
| | ELOC | Ours | ELOC | Ours | ELOC | Ours | ELOC | Ours |
| Tiny ImageNet | $\mathbf{2.94}$ | $1.91_{2.24}$ | $\mathbf{99.36}$ | $99.45_{0.67}$ | $\mathbf{24.53}$ | $18.68_{6.31}$ | $\mathbf{95.18}$ | $94.88_{1.75}$ |
| LSUN | $\mathbf{0.88}$ | $1.5_{1.80}$ | $\mathbf{99.7}$ | $99.61_{0.47}$ | 16.53 | $\mathbf{9.23}_{1.87}$ | 96.77 | $\mathbf{97.89}_{0.48}$ |
| Gaussian | $\mathbf{0.0}$ | $0.13_{0.20}$ | 99.58 | $\mathbf{99.95}_{0.08}$ | 98.26 | $\mathbf{0.72}_{0.79}$ | 93.04 | $\mathbf{99.65}_{0.39}$ |

vs. ODIN Liang et al. (2018)
(Model: Wide ResNet 28x10)

| $\mathcal{D}_{out}$ | $\mathcal{D}_{in}$: CIFAR-10 | | | | $\mathcal{D}_{in}$: CIFAR-100 | | | |
|---|---|---|---|---|---|---|---|---|
| | FPR95 $\downarrow$ | | AUROC $\uparrow$ | | FPR95 $\downarrow$ | | AUROC $\uparrow$ | |
| | ODIN | Ours | ODIN | Ours | ODIN | Ours | ODIN | Ours |
| Tiny ImageNet | 25.5 | $\mathbf{1.91}_{2.24}$ | 92.1 | $\mathbf{99.45}_{0.67}$ | 55.9 | $\mathbf{18.68}_{6.31}$ | 84.0 | $\mathbf{94.88}_{1.75}$ |
| LSUN | 17.6 | $\mathbf{1.5}_{1.80}$ | 95.4 | $\mathbf{99.61}_{0.47}$ | 56.5 | $\mathbf{9.23}_{1.87}$ | 86.0 | $\mathbf{97.89}_{0.48}$ |
| Gaussian | $\mathbf{0.0}$ | $0.13_{0.20}$ | $\mathbf{100.0}$ | $99.95_{0.08}$ | $\mathbf{1.0}$ | $0.72_{0.79}$ | 98.5 | $\mathbf{99.65}_{0.39}$ |

vs. Deep Mahalanobis Detector (MAH) Lee et al. (2018)
(Model: ResNet 34)

| $\mathcal{D}_{out}$ | $\mathcal{D}_{in}$: CIFAR-10 | | | | $\mathcal{D}_{in}$: CIFAR-100 | | | |
|---|---|---|---|---|---|---|---|---|
| | FPR95 $\downarrow$ | | AUROC $\uparrow$ | | FPR95 $\downarrow$ | | AUROC $\uparrow$ | |
| | MAH | Ours | MAH | Ours | MAH | Ours | MAH | Ours |
| SVHN | 24.2 | $\mathbf{1.89}_{0.78}$ | 95.5 | $\mathbf{99.49}_{0.17}$ | 58.1 | $\mathbf{41.31}_{8.01}$ | 84.4 | $\mathbf{86.85}_{2.38}$ |
| Tiny ImageNet | 4.5 | $\mathbf{0.36}_{0.15}$ | 99.0 | $\mathbf{99.88}_{0.04}$ | 29.7 | $\mathbf{12.10}_{1.22}$ | 87.9 | $\mathbf{97.14}_{0.29}$ |
| LSUN | 1.9 | $\mathbf{0.30}_{0.12}$ | 99.5 | $\mathbf{99.91}_{0.03}$ | 43.4 | $\mathbf{7.14}_{0.66}$ | 82.3 | $\mathbf{98.45}_{0.13}$ |

Table 1: Comparison of the extra class method (ours) with various other out-of-distribution detection methods when trained on CIFAR-10 and CIFAR-100 and tested on other datasets. All numbers from comparison methods are sourced from their respective original publications. For our method, we also report the standard deviation over five runs (indicated by the subscript), and treat the performance of other methods within one standard deviations as equivalent to ours. For fair comparison with the Mahalanobis detector (MAH) Lee et al. (2018), we use results when their method was not tuned separately on each OoD test set (Table 6 in Lee et al. (2018).

vs. OpenMax Bendale & Boult (2016)
(Model: ResNet 34)

| $\mathcal{D}_{out}$ | $\mathcal{D}_{in}$: CIFAR-10 | | | | $\mathcal{D}_{in}$: CIFAR-100 | | | |
|---|---|---|---|---|---|---|---|---|
| | FPR95 $\downarrow$ | | AUROC $\uparrow$ | | FPR95 $\downarrow$ | | AUROC $\uparrow$ | |
| | OpenMax | Ours | OpenMax | Ours | OpenMax | Ours | OpenMax | Ours |
| SVHN | $23.67_{2.06}$ | $\mathbf{1.89}_{0.78}$ | $90.72_{0.90}$ | $\mathbf{99.49}_{0.17}$ | $53.22_{7.52}$ | $\mathbf{41.31}_{8.01}$ | $80.88_{1.08}$ | $\mathbf{86.85}_{2.38}$ |
| Tiny ImageNet | $24.20_{9.11}$ | $\mathbf{0.36}_{0.15}$ | $93.39_{0.75}$ | $\mathbf{99.88}_{0.04}$ | $32.67_{5.21}$ | $\mathbf{12.10}_{1.22}$ | $81.22_{2.21}$ | $\mathbf{97.14}_{0.29}$ |
| LSUN | $18.68_{1.24}$ | $\mathbf{0.30}_{0.12}$ | $92.16_{1.82}$ | $\mathbf{99.91}_{0.03}$ | $30.21_{2.71}$ | $\mathbf{7.14}_{0.66}$ | $83.08_{2.16}$ | $\mathbf{98.45}_{0.13}$ |
| Places365 | $27.27_{2.77}$ | $\mathbf{0.84}_{0.28}$ | $90.72_{0.85}$ | $\mathbf{99.67}_{0.07}$ | $50.71_{1.25}$ | $\mathbf{30.54}_{2.26}$ | $81.13_{0.30}$ | $\mathbf{92.69}_{0.65}$ |
| Gaussian | $40.58_{22.18}$ | $\mathbf{0.04}_{0.02}$ | $84.74_{10.19}$ | $\mathbf{99.98}_{0.01}$ | $21.50_{11.73}$ | $\mathbf{1.66}_{1.76}$ | $89.37_{5.46}$ | $\mathbf{99.48}_{0.47}$ |

Table 2: DAC vs OpenMax. The OpenMax implementation was based on code available at https://github.com/abhijitbendale/OSDN and re-implemented by us in PyTorch Paszke et al. (2019).

vs. Outlier Exposure  Hendrycks et al. (2018) for NLP Classification
(Model: 2 layered GRU)

| $\mathcal{D}_{in}$ | $\mathcal{D}_{out}$ | FPR95 $\downarrow$ | | AUROC $\uparrow$ | |
|---|---|---|---|---|---|
| | | OE | Ours | OE | Ours |
| 20 Newsgroup | SNLI | 12.5 | $\mathbf{3.9}_{0.54}$ | 95.1 | $\mathbf{98.32}_{0.1}$ |
| | IMDB | 18.6 | $\mathbf{1.78}_{0.12}$ | 93.5 | $\mathbf{99.17}_{0.0}$ |
| | Multi30K | 3.2 | $\mathbf{0.8}_{0.13}$ | 97.3 | $\mathbf{99.52}_{0.04}$ |
| | WMT16 | 2.0 | $\mathbf{1.4}_{0.16}$ | 98.8 | $\mathbf{99.33}_{0.02}$ |
| | Yelp | 3.9 | $\mathbf{0.76}_{0.08}$ | 97.8 | $\mathbf{99.61}_{0.02}$ |
| TREC | SNLI | $\mathbf{4.2}$ | $12.0_{8.1}$ | $\mathbf{98.1}$ | $97.03_{1.08}$ |
| | IMDB | 0.6 | $\mathbf{0.0}_{0.0}$ | 99.4 | $\mathbf{99.99}_{0.0}$ |
| | Multi30K | $\mathbf{0.3}$ | $8.3_{3.4}$ | $\mathbf{99.7}$ | $97.56_{0.5}$ |
| | WMT16 | $\mathbf{0.2}$ | $4.67_{3.26}$ | 99.8 | $\mathbf{99.94}_{0.6}$ |
| | Yelp | 0.4 | $\mathbf{0.0}_{0.0}$ | 99.7 | $\mathbf{99.0}_{0.0}$ |
| SST | SNLI | 33.4 | $\mathbf{20.9}_{2.3}$ | 86.8 | $\mathbf{92.24}_{0.77}$ |
| | IMDB | 32.6 | $\mathbf{0.7}_{0.46}$ | 85.9 | $\mathbf{99.37}_{0.1}$ |
| | Multi30K | $\mathbf{33.0}$ | $70.9_{7.7}$ | $\mathbf{88.3}$ | $70.65_{4.9}$ |
| | WMT16 | $\mathbf{17.1}$ | $31.6_{5.9}$ | $\mathbf{92.9}$ | $90.64_{1.7}$ |
| | Yelp | 11.3 | $\mathbf{0.06}_{0.00}$ | 92.7 | $\mathbf{99.67}_{0.08}$ |

Table 3: DAC vs OE for NLP Classification task. OE implementation was based on code available at
`https://github.com/hendrycks/outlier-exposure`

one expects that such methods will cause the DNN to predict with less confidence when presented with inputs from a different distribution or from novel categories; we compare against the following methods:

- **Softmax Thresholding** This is the simplest baseline, where OoD samples are detected by thresholding on the winning softmax score; scores falling *below* a threshold are rejected.

- **Entropy Thresholding** Another simple baseline, where OoD samples are rejected if the Shannon entropy calculated over the softmax posteriors is *above* a certain threshold.

- **MonteCarlo Dropout** A Bayesian inspired approach proposed in Gal & Ghahramani (2016) for improving the predictive uncertainty for deep learning. We found a dropout probability of $p = 0.5$ to perform well, and use 100 forward passes per sample during the prediction.

- **Temperature Scaling**, which improves DNN calibration as described in Guo et al. (2017). The scaling temperature $T$ is tuned on a held-out subset of the validation set of the in-distribution data.

- **Mixup** As shown in Thulasidasan et al. (2019), Mixup can be an effective OoD detector, so we also use this as one of our baselines.

- **Deep Ensembles** which was introduced in Lakshminarayanan et al. (2017) for improving uncertainty estimates for both classification and regression. In this approach, multiple versions of the same model are trained using different random initializations, and while training, adversarial samples are generated to improve model robustness. We use an ensemble size of 5 as suggested in their paper.

- **SWAG**, as described in Maddox et al. (2019), which is a Bayesian approach to deep learning and exploits the fact that SGD itself can be viewed as approximate Bayesian inference Mandt et al. (2017). We use an ensemble size of 30 as proposed in the original paper.

### 3.3.1 RESULTS

Detailed results are shown in Table 4, where the best performing method for each metric is shown in bold. The DAC is the only method in this set of experiments that uses an augmented dataset, and as is clearly evident from the results, this confers a significant advantage over the other methods in most cases. Calibration methods like temperature scaling, while producing well calibrated scores on in-distribution data, end up reducing the confidence on in-distribution data as well, and thus losing discriminative power between the two types of data. We also note here that many of the methods listed in the table, like temperature scaling and deep ensembles, can be combined with the

abstention approach. Indeed, the addition of an extra abstention class and training with OoD data is compatible with most uncertainty modeling techniques in deep learning; we leave the exploration of such combination approaches for future work.

| $\mathcal{D}_{out}$ | $\mathcal{D}_{in}$: CIFAR-100 AUROC ↑ | | | | | | | |
| | Softmax | Entropy | Monte Carlo Dropout | Temp Scaling | Mixup | Deep Ensemble | SWAG | DAC (Ours) |
|---|---|---|---|---|---|---|---|---|
| LSUN | $87.86_{0.54}$ | $89.53_{0.68}$ | $84.35_{2.99}$ | $87.75_{0.70}$ | $86.03_{1.65}$ | $89.36_{0.23}$ | $83.87_{2.59}$ | $\mathbf{98.73}_{0.10}$ |
| Places-365 | $79.77_{0.73}$ | $80.34_{0.83}$ | $80.21_{1.06}$ | $79.53_{1.08}$ | $82.09_{0.98}$ | $82.69_{0.27}$ | $83.37_{0.42}$ | $\mathbf{93.39}_{0.59}$ |
| Gaussian | $80.76_{6.52}$ | $80.46_{8.50}$ | $68.40_{23.29}$ | $80.52_{6.51}$ | $92.13_{6.08}$ | $80.27_{5.01}$ | $91.80_{6.94}$ | $\mathbf{99.69}_{0.28}$ |
| SVHN | $78.45_{3.97}$ | $79.43_{4.30}$ | $81.04_{2.46}$ | $78.25_{3.99}$ | $79.15_{2.52}$ | $82.42_{0.22}$ | $80.45_{1.67}$ | $\mathbf{87.74}_{2.34}$ |
| Tiny ImageNet | $86.63_{0.92}$ | $88.00_{1.22}$ | $82.07_{4.44}$ | $86.49_{1.14}$ | $84.57_{1.61}$ | $86.64_{0.23}$ | $80.90_{2.37}$ | $\mathbf{97.60}_{0.25}$ |

| $\mathcal{D}_{out}$ | $\mathcal{D}_{in}$: CIFAR-100 FPR95 ↓ | | | | | | | |
| | Softmax | Entropy | Monte Carlo Dropout | Temp Scaling | Mixup | Deep Ensemble | SWAG | DAC (Ours) |
|---|---|---|---|---|---|---|---|---|
| LSUN | $35.64_{1.65}$ | $34.52_{1.82}$ | $42.93_{4.60}$ | $35.64_{1.65}$ | $34.52_{1.82}$ | $32.48_{0.37}$ | $40.50_{4.09}$ | $\mathbf{5.41}_{0.58}$ |
| Places-365 | $55.19_{1.36}$ | $55.81_{1.40}$ | $56.50_{1.33}$ | $55.19_{1.36}$ | $55.81_{1.40}$ | $46.55_{0.81}$ | $47.76_{0.91}$ | $\mathbf{29.59}_{2.48}$ |
| Gaussian | $33.70_{8.14}$ | $32.82_{10.19}$ | $41.34_{23.76}$ | $33.70_{8.14}$ | $32.82_{10.19}$ | $23.85_{5.07}$ | $13.61_{10.38}$ | $\mathbf{0.89}_{0.96}$ |
| SVHN | $55.28_{10.30}$ | $55.16_{10.67}$ | $50.08_{4.49}$ | $55.28_{10.30}$ | $55.16_{10.67}$ | $47.57_{0.53}$ | $49.43_{3.63}$ | $\mathbf{40.44}_{8.31}$ |
| Tiny ImageNet | $37.27_{1.04}$ | $36.65_{1.25}$ | $47.10_{6.05}$ | $37.27_{1.04}$ | $36.65_{1.25}$ | $38.58_{0.51}$ | $45.05_{4.08}$ | $\mathbf{10.15}_{1.09}$ |

| $\mathcal{D}_{out}$ | $\mathcal{D}_{in}$: CIFAR-10 AUROC ↑ | | | | | | | |
| | Softmax | Entropy | Monte Carlo Dropout | Temp Scaling | Mixup | Deep Ensemble | SWAG | DAC |
|---|---|---|---|---|---|---|---|---|
| LSUN | $93.45_{0.36}$ | $93.84_{0.37}$ | $91.56_{1.77}$ | $93.99_{0.89}$ | $96.58_{1.38}$ | $93.44_{0.19}$ | $96.28_{0.06}$ | $\mathbf{99.92}_{0.02}$ |
| Places-365 | $91.70_{0.56}$ | $92.08_{0.58}$ | $89.29_{0.72}$ | $92.39_{1.00}$ | $96.38_{0.87}$ | $94.78_{0.23}$ | $96.07_{0.63}$ | $\mathbf{100.00}_{0.00}$ |
| Gaussian | $79.39_{13.06}$ | $79.22_{13.24}$ | $94.26_{2.74}$ | $80.40_{9.22}$ | $95.62_{3.69}$ | $95.82_{1.30}$ | $96.51_{0.52}$ | $\mathbf{100.00}_{0.00}$ |
| SVHN | $91.38_{1.03}$ | $91.70_{1.11}$ | $83.56_{3.85}$ | $91.99_{0.34}$ | $92.62_{2.96}$ | $90.58_{0.20}$ | $96.12_{1.19}$ | $\mathbf{99.50}_{0.16}$ |
| Tiny ImageNet | $91.98_{1.85}$ | $92.31_{1.88}$ | $88.84_{5.19}$ | $92.59_{1.47}$ | $95.67_{2.02}$ | $94.07_{0.29}$ | $95.65_{0.13}$ | $\mathbf{99.88}_{0.04}$ |

| $\mathcal{D}_{out}$ | $\mathcal{D}_{in}$: CIFAR-10 FPR95 ↓ | | | | | | | |
| | Softmax | Entropy | Monte Carlo Dropout | Temp Scaling | Mixup | Deep Ensemble | SWAG | DAC |
|---|---|---|---|---|---|---|---|---|
| LSUN | $19.15_{1.38}$ | $18.93_{1.30}$ | $25.86_{4.46}$ | $18.76_{2.65}$ | $12.47_{4.88}$ | $19.36_{0.54}$ | $12.71_{0.02}$ | $\mathbf{0.27}_{0.12}$ |
| Places-365 | $26.49_{3.65}$ | $26.42_{3.79}$ | $33.65_{2.31}$ | $26.37_{6.27}$ | $14.77_{4.62}$ | $15.79_{0.39}$ | $12.80_{1.67}$ | $\mathbf{0.00}_{0.00}$ |
| Gaussian | $52.84_{22.15}$ | $53.52_{21.98}$ | $12.63_{4.24}$ | $55.91_{18.40}$ | $10.96_{11.42}$ | $9.15_{1.36}$ | $9.55_{0.52}$ | $\mathbf{0.00}_{0.00}$ |
| SVHN | $24.64_{2.41}$ | $24.67_{2.49}$ | $53.32_{14.76}$ | $25.06_{3.46}$ | $29.71_{12.57}$ | $24.99_{0.65}$ | $13.04_{4.05}$ | $\mathbf{1.78}_{0.76}$ |
| Tiny ImageNet | $26.21_{10.40}$ | $26.24_{10.66}$ | $33.25_{15.67}$ | $24.96_{5.58}$ | $14.84_{6.12}$ | $17.03_{0.58}$ | $12.07_{0.14}$ | $\mathbf{0.33}_{0.14}$ |

Table 4: The performance of DAC as an OoD detector, evaluated on various metrics and compared against competing baselines. All experiments used the ResNet-34 architecture, except for MC Dropout, in which case we used the WideResNet 28x10 network. ↑ and ↓ indicate that higher and lower values are better, respectively. Best performing methods (ignoring statistically insignificant differences) on each metric are in bold.

## 4 CONCLUSION

We presented a simple, but highly effective method for open set and out-of-distribution detection that clearly demonstrated the efficacy of using an extra abstention class and augmenting the training set with outliers. While previous work has shown the efficacy of outlier exposure Hendrycks et al. (2018), here we demonstrated an alternative approach for exploiting outlier data that further improves upon existing methods, while also being simpler to implement compared to many of the other methods. The ease of implementation, absence of additional hyperparameter tuning and computational efficiency during testing makes this a very viable approach for improving out-of-distribution and novel category detection in real-world deployments; we hope that this will also serve as an effective baseline for comparing future work in this domain.

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
