# OpenReview forum: "A Simple and Effective  Baseline for Out-of-Distribution Detection using Abstention"
_ICLR.cc/2021/Conference — Reject_

### Official Review · AnonReviewer1 · 2020-10-28
**The observation is interesting, but more experiments are required**

**Rating:** 4
**Confidence:** 4

**Review:**

- Summary:
This paper shows that introducing an abstention class for out-of-distribution (OOD) works well for detecting it when the in-distribution dataset is CIFAR and TinyImageNet is available during training as an OOD dataset.

- Reasons for score:
1. The proposed setting with a large OOD dataset has already been proposed by [Hendrycks et al.], and the proposed method has been experimented in [Lee et al. (a)] and [Dhamija et al.], so the technical novelty of this work is limited.
2. The experiments are not thoroughly conducted. The only value I can find in this paper is the empirical observation in a limited condition. More specifically, this paper found that adding an abstention class is better than prior methods when the in-distribution dataset is CIFAR and TinyImageNet is available during training as an OOD dataset. I am not sure this observation is consistent in other settings, so I recommend to conduct more thorough experiments, as done in [Hendrycks et al.]. Again, [Hendrycks et al.] considered a similar setting, but they proved the effectiveness of their method in image and natural language domains, with 7 in-distribution datasets and 3 large OOD datasets. However, even with more experiments, I am not sure this work is significant enough for publication in ICLR, because of the lack of novelty in both the experimental scenario and method.
3. The comparison is unfair. Performances of prior methods are borrowed from original works, but they are mostly experimented in settings different from this paper. In particular, some works like [Lee et al. (b)] and [Hsu et al.] considered training/validating without OOD data, because it is hard to assume to have a large OOD dataset covering all possible OOD in practice.

- Minor Comments:
4. Citation format issue: you can use \citet for noun and \citep for adverb.

[Lee et al. (a)] Training confidence-calibrated classifiers for detecting out-of-distribution samples. In ICLR, 2018.

[Lee et al. (b)] A simple unified framework for detecting out-of-distribution samples and adversarial attacks. In NeurIPS, 2018.

[Dhamija et al.] Reducing network agnostophobia. In NeurIPS, 2018.

[Hendrycks et al.] Deep anomaly detection with outlier exposure. In ICLR, 2019.

[Hsu et al.] Generalized ODIN: Detecting Out-of-distribution Image without Learning from Out-of-distribution Data. In CVPR, 2020.

**After rebuttal**

I'd like to thank authors for their efforts to address my concerns. I didn't change my initial rating, due to the two main concerns below:

(1) To me, the main argument of this paper sounds "when a large (and maybe diverse) OOD is given, adding an OOD class to the classifier is better than baselines." Since the large OOD setting has already been proposed by [Hendrycks et al.], the only contribution of this work is on the empirical observation that the proposed method is better than baselines. While the observation is interesting, I think the contribution is not enough as a full ICLR paper at this point.

During the rebuttal period, R3 corrected it that "the main question investigated by the paper is how to best use the outlier exposure set," and this sounds better. However, authors didn't emphasize the setting but their method, such that their main argument is (if they intended to say as like what R3 understood) misleading. Training with a large OOD dataset like [Hendrycks et al.] is not common, and the observation in this paper is limited to this setting. However, the only statement about the setting I could find in the intro is that "as in Hendrycks et al. (2018), we uses additional samples of real images and text from non-overlapping categories to train the model to abstain, ..." i.e., rather than elaborating/emphasizing the setting (together with their method), they just cited a prior work.

In short, I recommend authors to rewrite abstract/intro as suggested by R3, to properly emphasize their contribution.

(2) The comparison is unfair, as authors didn't re-evaluate baselines in the same setting (they had to make it the same as much as possible) but just pasted numbers from original papers. Even the comparison with the closest prior work [Hendrycks et al.] is unfair, as the prior work fine-tuned the model while the proposed method trained the model from scratch.

Regarding the performance of similar methods evaluated in [Lee et al. (a)] and [Dhamija et al.], I think the main reason why they didn't get the same observation is on the size of OOD dataset, i.e., they didn't train their model with a large OOD dataset like [Hendrycks et al.] or this work. As this work claims, it might be true that when a large OOD dataset is available, adding an OOD class to the classifier is simply good enough.

---

> ### Author Response · Authors · 2020-11-25
> **Response to AnonReviewer 1**
>
> Thank you for your comments. The main objection here seems to be that the method might only work when $D_{in}^{train}$ is CIFAR-10 or 100, when coupled with Tiny Images as $D_{out}^{train}$.
>
> The paper focussed on  the results for this combination to make it comparable against other methods. However, the strong performance is not limited to these datasets. For example, in one set of experiments, we used Tiny ImageNet (not to be confused with  TinyImages) [1] as our $D_{train}^{in}$, and an unlabeled portion of STL-10 as  $D_{train}^{out}$ and compared with various uncertainty based methods. As you can see from the results below, the performance improvements are so significant that there really is no comparison; we do not report these in the paper, since most papers report results on CIFAR,  as testing for OoD recognition on CIFAR is a more challenging task.
>
> +---------------+------------+---------+---------+------------+--------------+-------+---------------+-------+----------+--------------+----------+
>
> |    id_data |  ood_data          | Softmax | Entropy | MC Dropout | Temp Scaling | Mixup | Deep Ensemble | SWAG  |   **DAC**    |
>
> +---------------+------------+---------+---------+------------+--------------+-------+---------------+-------+----------+--------------+----------+
>
> | Tiny ImageNet | ImageNet   |   73.20 |  73.99  |      72.38 |        82.35 | 73.37 |         72.55 | 71.83 |  **99.29**   |
>
> | Tiny ImageNet | Places-500 |   75.85 |   76.76 |      75.89 |        85.39 | 76.98 |         74.40 | 72.58 | **100.00**   |
>
> | Tiny ImageNet | Gaussian   |   50.27 |   51.72 |      52.32 |        64.60 | 55.21 |         66.04 | 57.39 | **100.00**   |
>
> +---------------+------------+---------+---------+------------+--------------+-------+---------------+-------+----------+
>
> Further there is an entire table of results in the submitted version  devoted to results on NLP datasets (Table 3) where we are able to achieve strong OoD detection performance  (AUROC > 0.97 in most cases).
>
> In summary, our method works  very well under multiple data scenarios, and performs strongly against a wide variety of baseline methods.
>
> Regarding fairness of comparison: you are right that the methods like Lee et al[b]  and Hsu et al do not use  OoD data; if such data is not available then our method is  not applicable.  While one cannot make an assumption regarding the availability of OoD data for training, in many situations such data is available, or can be easily collected.
>
>  Further, the results in our paper show that even when the distribution of $D_{out}^{train}$ is different from $D_{out}^{test}$, the presence of an extra class and the model’s exposure to  examples different from $D_{in}$ confers significant performance benefits.
>
> Regarding novelty:
>
> We make the distinction between Hendrycks et al's work and ours  in the paper. While both methods use natural outliers, their method uses an entropy regularizer, and our method uses an extra class; we believe the latter is simpler to implement.
>
> Regarding Lee et al [a]:  Appendix E of their paper discusses the results of adding an extra class. A couple of points worth discussing: in their method, $D_{out}^{train}$ is the output of a GAN, whereas our $D_{out}^{train}$ is a naturally occurring dataset, and in many cases simpler to obtain than training a GAN to convergence.  But more importantly,  in Table 6 of said Appendix, for the extra class method, AUROCs of less than 0.5 (sometimes significantly less than)  are reported, which makes interpretation somewhat problematic.
>
> And finally w.r.t Dhamija et al, even though their method uses natural outliers, we hope you agree that their formulation — the objectosphere loss combined with entropy-regularized open-set loss —  is much more involved; in contrast, ours uses nothing more than vanilla cross-entropy.
>
> References;
>
> 1. Tiny ImageNet https://www.kaggle.com/c/tiny-imagenet

---

> > ### Comment · AnonReviewer1 · 2020-11-25
> > **Response from AnonReviewer 1**
> >
> > Thank you for the answer. Here are my responses:
> >
> > I understand that you followed the experimental setting of OE [Hendrycks et al.], which uses TinyImages and Wikitext2 as OOD datasets for training in image and NLP domains, respectively. Then, I would say the main contribution of this work is on the empirical observation that "when a large OOD is available like the setting in OE, then adding an OOD class to the classifier is better than methods proposed in OE." However, again, the comparison with OE seems not fair, as numbers are just copied from the original paper.
> >
> > Note that OE fine-tuned a pre-trained network in their main results, and claimed that training from scratch results in better performance. However, in this work, it seems the provided code is training from scratch, and I couldn't find any word "fine-tuning" or something similar in the manuscript. So, for a fair comparison, OE should be re-evaluated by training from scratch, or the proposed method should be evaluated by fine-tuning the network. Similarly, for other compared methods, you should re-evaluate them in the same condition.
> >
> > Thank you for providing more results. However, in the results you newly provided, TinyImageNet is a subset of ImageNet, so please make sure your ImageNet (I believe you are referring to ILSVRC 2012) has only 800 classes. Even in this case, the result looks not convincing, because ImageNet contains many fine-grained classes. "Detecting unseen and fine-grained classes almost perfectly" seems unrealistic.
> >
> > Regarding the performance of similar methods evaluated in [Lee et al. (a)] and [Dhamija et al.], I think the main reason why they didn't get the same observation is on the size of OOD dataset, i.e., they didn't train their model with a large OOD dataset. As this work claims, it might be true that when a large OOD dataset is available, adding an OOD class to the classifier is simply good enough. So I would say, the observation in this work is interesting, but more experiments had to be conducted for a fair comparison.

---

### Official Review · AnonReviewer3 · 2020-10-29
**Sensible idea, but needs more complete experiments**

**Rating:** 5
**Confidence:** 4

**Review:**

**Update after author response:** I have read the other reviewer's comments. My take is that at a high level the contribution of this paper is above the bar of ICLR, but the experiments aren't controlled enough so I vote for a weak reject.

Hendrycks et al propose encouraging high entropy predictions on the outlier exposure set, instead of classifying them into a reject class. Going further, Hendrycks et al claim "but we find that even with OE, classifiers with the reject option... are not as competitive". As I understand, Hendrycks et al are basically saying the (simpler) method in this paper does not work as well - but Hendrycks et al don't provide experimental evidence for this claim, it's merely stated. The original paper might in fact discourage practitioners from trying this approach, and instead using the high entropy approach. In fact, this was a question in my mind when I read Hendrycks et al last year.

Instead, this paper seems to show stronger results, with a more intuitive and simpler method (just classify the outlier exposure set into a separate class) that Hendrycks et al suggest doesn't work as well. Conceptually, the approach in Hendrycks et al also seems more brittle and there are distinctions between these two methods (e.g. see Vernekar et al 2019, “Analysis of Confident-Classifiers for Out-of-distribution Detection”).

So what's the potential practical impact? If this paper didn't exist, I suspect practitioners would use the method in Hendrycks et al, and not try the reject class method, because of that paper's claims. But with this paper, practitioners might use this method or try both, which seems like a good impact.

So barring problems with the experimental setup, I'd give the paper a 7 / accept, and so I'd encourage the authors to continue on with this work.

To me the decision hinges on the quality of the comparisons. I am inclined to agree with R1 on the quality of comparisons. Taking a closer look at their paper, they have no detailed discussion about the hyperparameters and experimental setup, which is key when the main contribution is a fine-grained comparison with Hendrycks et al. R1 raises a question about fine-tuning vs trained from scratch, and it does look like this paper trains from scratch whereas outlier exposure fine-tunes. The outlier exposure set is also different. While it is a smaller set in this paper, Hendrycks et al say "experiments in this paper often used around 1% of the images in the 80 Million Tiny Images dataset since we only briefly fine-tuned the models.".

Overall, I agree that the best thing would probably be for them to do a more careful  and controlled comparison, include all these details, and submit to the next conference. In my review I did mention that their comparisons were unclear, but they didn't take the chance to update their paper and misleadingly responded that "The OE method is closest to ours, so we are able to match their training regimen well", but as R1 points out there seem to be salient differences.

#########################################################################

Summary:

This paper tackles the problem of out of distribution detection. Concretely, they have an in-distribution set of inputs D_in and the goal is to reject inputs from an unknown set D_out as out of distribution, while accepting inputs from D_in as in-distribution. Like in the outlier exposure paper, they assume a known set of proxy out of domain distribution inputs \tilde{D}_out. The approach is to classify inputs in D_in into one of K classes, while classifying inputs from \tilde{D}_out into a “K+1”-th class. On a few vision and NLP datasets, they show better performance than outlier exposure and other methods.

#########################################################################

Reasons for score:

The main technical difference compared to outlier exposure is that they classify inputs from \tilde{D}_out into a K+1-th class, whereas outlier exposure enforces that the prediction over the K classes have high entropy if the inputs are from \tilde{D}_out. 1. One main concern is that they have other (hidden) differences from Hendrycks et al, for example I looked at their code and they use Mixup and a different \tilde{D}_out, so it is unclear if the gain is actually coming from the K+1-th class, or e.g. more modern data augmentation practices. They also don’t test on the most challenging image datasets from Hendrycks et al, in particular D_in = CIFAR-100, D_out = CIFAR-10 (they have distinct sets of classes). 2. This method was also proposed by Vernekar et al, “Analysis of Confident-Classifiers for Out-of-distribution Detection”.

I generally believe that this is the right thing to do, and seems simpler and more sound than enforcing high entropy, and Vernekar et al only have toy experiments. However, since the contribution is a small change to OE, to make this complete I believe they should have more complete experiments. In other words, I’m holding them to a higher bar experimentally, than if say their idea was novel or they had conceptual insights into the problem, because without more complete experiments the contribution to the community is limited. Note that unlike most works on OOD detection, they use more data, \tilde{D}_out, similar to outlier exposure. This is completely fine, but that's why I think simply having SOTA results is not quite enough to push this over the bar since there's only about 1 paper on this precise setup.

On the plus side, this work should at least convince practitioners to try both methods, K+1-th class and enforcing high entropy.

#########################################################################

Pros:

- Simple idea, and does better than competing methods on most baselines.

- Combination of vision and NLP datasets.

- Well written, clear and simple.

#########################################################################

Cons:

- I looked at this paper’s code, and Mixup is an additional difference compared to Hendrycks et al’s outlier exposure. There could be other differences I did not spot (e.g. training procedure? Optimization method). It’s great that the overall method does better, but I’d like to see at experiments on some datasets investigating what happens if you use the same procedure as OE except K+1-th class instead of entropy.

- Results on CIFAR-10 -> CIFAR-100, and CIFAR-100 -> CIFAR-10 would be good. Since the main contribution is experimental, it would be particularly compelling if a couple more datasets were added. E.g. the original OE paper is substantially more comprehensive. E.g. one combo could be Places365 -> ImageNet.

- No conceptual reason for why the K+1-th method does better (I am still happy to vote to accept without this, but this would make the paper more compelling). The earlier work by Vernekar et all doesn’t quite explain things for high dimensional data like images.


#########################################################################

Questions and things to improve:

If the comparisons to Hendrycks can be re-run with matching training procedure, and they show results on a few more cases (e.g. CIFAR-10 -> CIFAR-100, and CIFAR-100 -> CIFAR-10), I would seriously consider leaning towards an accept.

#########################################################################

---

> ### Author Response · Authors · 2020-11-15
> **Mixup is not used while training the abstaining classifier**
>
> Hi, thanks for your comments.  We would like to provide a quick clarification: mixup is not used for training the proposed model. The reason you see it in the code is we use the same training script for running the mixup experiments, since it's also one of the methods we compare against.
>
> Comments regarding other points you raised will be posted here soon.
>
> Regards,
> Authors.

---

> > ### Comment · AnonReviewer3 · 2020-11-23
> > **Looking forward to other responses**
> >
> > Cool, looking forward to the responses to the rest of the questions.

---

> > > ### Author Response · Authors · 2020-11-25
> > > **Response to AnonReviewer 3**
> > >
> > > We appreciate your comments and actionable suggestions. Responses below:
> > >
> > > > I looked at this paper’s code, and Mixup is an additional difference compared to Hendrycks et al’s outlier exposure. There could be other differences I did not spot (e.g. training procedure? Optimization method). It’s great that the overall method does better, but I’d like to see at experiments on some datasets investigating what happens if you use the same procedure as OE except K+1-th class instead of entropy.
> > >
> > > For comparing with Outlier Exposure (OE), we used the same network architecture (Wide ResNet 40x2), optimization method (SGD with Nesterov), initial learning rate (0.1) and weight decay (5e-4) that Hendrycks et al  use in their CIFAR-10/100 experiments. As clarified in the earlier comment, mixup is not used.
> > >
> > > The OE method is closest to ours, so we are able to match their training regimen well, but this is also generally true for other OoD methods we compare  against. Some of the other methods (like Mahalanobis) are very different, but we have used the same architectures and optimization method (as reported in those respective papers.)
> > >
> > > > Results on CIFAR-10 -> CIFAR-100, and CIFAR-100 -> CIFAR-10 would be good. Since the main contribution is experimental, it would be particularly compelling if a couple more datasets were added. E.g. the original OE paper is substantially more comprehensive. E.g. one combo could be Places365 -> ImageNet.
> > >
> > > CIFAR 10 vs 100 results compared to OE method given below.  We consider the performance to be statistically equivalent,  and as you and others have pointed out, this is a  challenging case.
> > >
> > > —————————————————————————————————————————
> > >
> > > D_in | D_out | AUROC (ours) | AUROC (OE)| FPR 95 (ours)| FPR 95 (OE)
> > >
> > > —————————————————————————————————————————
> > >
> > > CIFAR-10 | CIFAR-100 | 93.07+/-0.73 | 93.3 | 29.42+/-2.72 | 28.0
> > >
> > > CIFAR-100| CIFAR-10 | 75.17+/-1.56 | 75.7 | 63.00+/-2.81 | 62.1
> > >
> > > —————————————————————————————————————————
> > >
> > > While the OE paper does indeed compare against additional datasets, we do compare against many more methods (12 different methods for OoD detection and uncertainty estimation). All the uncertainty-based methods (Table 4) were re-implemented and evaluated under the same settings. One of the take-aways in this paper is that methods explicitly formulated for OoD do better than uncertainty-based methods for OoD detection.
> > >
> > > > No conceptual reason for why the K+1-th method does better (I am still happy to vote to accept without this, but this would make the paper more compelling). The earlier work by Vernekar et all doesn’t quite explain things for high dimensional data like images.
> > >
> > >  In our method, the classifier learns to map features that are not in $D_{in}^{train}$, but present in $D_{out}^{train}$, to the K+1st class i.e., it learns a more targeted mapping than in methods like OE and Lee et al, where the loss only cares about enforcing high entropy for the outliers. We observe that this targeted mapping of features to a specific outlier class performs better in certain situations. Ultimately all these methods will depend on how good $D_{out}^{train}$ is, but we think our method is simpler.
> > >
> > > References:
> > >
> > > Hendrycks et al. Deep Anomaly Detection with Outlier Exposure, ICLR '19
> > >
> > > Lee et al  Training confidence-calibrated classifiers for detecting out-of-distribution samples, ICLR '18.

---

> > > > ### Comment · AnonReviewer3 · 2020-11-25
> > > > **Thanks for the clarification, updated to weak accept**
> > > >
> > > > I've updated to a weak accept for now. For any remaining updates, I'm mainly curious about the differences between this work and Mohseni et al (e.g. is it just more datasets and a smaller OOD dataset), or are there other differences.

---

### Official Review · AnonReviewer4 · 2020-10-29
**using a reject class for outlier detection**

**Rating:** 4
**Confidence:** 5

**Review:**

-- Summary:
This paper presents experiments and results from using a reject-class in multi-class classification for the auxiliary task of out-of-distribution (ood) sample detection. Training requires unlabeled ood data disjoint from the training set.


--  Review:
1- The idea of using auxiliary task or reject-class for ood detection has been well explored in the past. Examples include Neal et al. [1] mentioned in the paper and recently by Mohseni et al [2] which is missing from the reference. This also has been used to improve representation learning in Zhang and LeCun [3].

2- Previous work accepted and recognized MSP score (with not additional data or hyperparameter) as the baseline for ood detection. What is your justification to propose your reject-class technique as the new baseline?

3- A challenging test case for ood detection is training and testing on two similar datasets such as cifar10 and cifar100. The cifar10 as D-in for train and cifar100 as D-out for test (and vice versa) is a very common example. I wish we could see results on that.

-- Strengths:
- The ood detection problem is an important and interesting topic
- Authors reviewed results and compared the proposed ood detection technique with multiple uncertainty-based techniques.


-- Weaknesses:
- Lack of novel contributions.
- Experiments and results are limited. More recent papers studied the use of self-supervised learning (e.g. [4, 5] ) and contrastive learning (e.g., [6, 7,8]) to improve ood detection.


[1] Neal, Lawrence, et al. "Open set learning with counterfactual images." Proceedings of the European Conference on Computer Vision (ECCV). 2018.'
[2] Mohseni, Sina, et al. "Self-Supervised Learning for Generalizable Out-of-Distribution Detection." AAAI. 2020.
[3] Zhang, Xiang, and Yann LeCun. "Universum prescription: Regularization using unlabeled data." Thirty-First AAAI Conference on Artificial Intelligence. 2017.
[4] Hendrycks, D., Mazeika, M., Kadavath, S., & Song, D. (2019). Using self-supervised learning can improve model robustness and uncertainty. In Advances in Neural Information Processing Systems (pp. 15663-15674).
[5] Golan, Izhak, and Ran El-Yaniv. "Deep anomaly detection using geometric transformations." Advances in Neural Information Processing Systems. 2018.
[6] Tack, J., Mo, S., Jeong, J., & Shin, J. (2020). Csi: Novelty detection via contrastive learning on distributionally shifted instances. arXiv preprint arXiv:2007.08176.
[7] Winkens, J., Bunel, R., Roy, A. G., Stanforth, R., Natarajan, V., Ledsam, J. R., ... & Cemgil, T. (2020). Contrastive training for improved out-of-distribution detection. arXiv preprint arXiv:2007.05566.
[8] Liu, Hao, and Pieter Abbeel. "Hybrid discriminative-generative training via contrastive learning." arXiv preprint arXiv:2007.09070 (2020).

---

> ### Comment · AnonReviewer3 · 2020-11-23
> **Outlier exposure much stronger than baseline**
>
> I'll let the authors answer the rest of it, but outlier exposure (https://arxiv.org/abs/1812.04606) and MAH are much better than the MSP baseline (e.g. https://arxiv.org/pdf/1610.02136.pdf), so it seems fine to compare with outlier exposure and MAH. Outlier exposure uses extra data.

---

> ### Author Response · Authors · 2020-11-24
> **Response to AnonReviewer 4**
>
> Thank you for your feedback, suggestions, and pointers to very recent,  related  work.
>
> Responses below.
>
> > The idea of using auxiliary task or reject-class for ood detection has been well explored in the past. Examples include Neal et al. [1] mentioned in the paper and recently by Mohseni et al [2] which is missing from the reference. This also has been used to improve representation learning in Zhang and LeCun [3].
>
> You are right that this is not the first paper to look at this idea per se, but Neal’s paper uses a GAN to generate OoD data, and ours uses a readily available, but uncurated, set, and is thus a much more simpler method. Mohseni’s work is very recent, and thank you for bringing this to our attention. It is indeed very similar (both in terms of methods and datasets), but to be fair, they only compare their performance to maximum softmax  and outlier exposure. In contrast, we have compared our approach to  12 different methods, both from the OoD literature as well as the uncertainty literature. In that sense, our results are much more comprehensive.
>
> There  is also one additional notable difference with Mohseni’s work: they expose their model to all 80 million tiny images, while we only used a very small,  randomly sampled fraction (100K) of the available data.
>
> Zhang and LeCun is related, though their goal was not OoD detection. However, we will certainly add  references to both of these works; Mohseni’s  is especially very relevant.
>
> >  Previous work accepted and recognized MSP score (with not additional data or hyperparameter) as the baseline for ood detection. What is your justification to propose your reject-class technique as the new baseline?
>
> Very fair question, and we justify as follows: Our view of what constitutes an effective baseline is that it should be 1.) relatively simple, and 2.) a simple method with strong performance is an even better baseline. While MSP is certainly simple, and might often be a sufficient baseline, in many cases one does have access to OoD data. It is in  such cases that we believe our method can serve as a strong baseline.
>
> As we have shown, in such cases, our method is not only much better than MSP, but is often the best performing method  compared to much more complicated approaches.  Also, the fact that our method does not bother with a curation step for the OoD training data makes this even more practical. To summarize, we believe that for practitioners with access to OoD data, this  method offers both simplicity and performance.
>
> >  A challenging test case for ood detection is training and testing on two similar datasets such as cifar10 and cifar100. The cifar10 as D-in for train and cifar100 as D-out for test (and vice versa) is a very common example. I wish we could see results on that.
>
> CIFAR 10 vs 100 given below, and compared to OE. We consider the performance to be statistically equivalent,  and as you  point out, this is a  challenging case.
>
> —————————————————————————————————————————
>
> D_in          | D_out        | AUROC (ours)  | AUROC (OE)| FPR 95 (ours)| FPR 95 (OE)
>
> —————————————————————————————————————————
>
> CIFAR-10  | CIFAR-100	| 93.07+/-0.73    |   93.3              | 29.42+/-2.72  | 28.0
>
> CIFAR-100| CIFAR-10	| 75.17+/-1.56    |   75.7              | 63.00+/-2.81  | 62.1
>
> —————————————————————————————————————————
>
> From a practitioner’s point of view, one could argue that in this case, our approach is easier to implement as it does not introduce any additional hyperparameters into the loss function (as done in OE). There are also results in Mohseni’s work where they show better performance, but as discussed before, their method uses the entire 80 million image dataset of TinyImages, so numbers are not directly comparable.

---

> > ### Comment · AnonReviewer3 · 2020-11-25
> > **Comparison with Mohseni**
> >
> > I hadn't seen Mohseni's work, but just to clarify is the only difference that this work has more datasets, and the OOD dataset? Curious what R4 and the authors think about this.

---

### Official Review · AnonReviewer2 · 2020-10-31
**Simple idea for an OOD baseline, but experiments need more analysis**

**Rating:** 6
**Confidence:** 3

**Review:**

I read this paper with great interest. The authors propose an easy-to-understand, easy-to-implement baseline method for detecting when inputs to a ML model is out of distribution. The method involves augmenting the training dataset with  an out of distribution dataset and adding an additional class in the classification layer for out of distribution. The paper describes several experiments—some in computer vision, some in NLP—and then compares them to other OOD techniques. The results are comparable to other techniques, although the proposed technique is definitely simpler.

I believe that this is a direction worth exploring; however, I do feel that the authors could have provided some more insights into the following decisions:
1. What are the desired characteristics of the OOD dataset used to augment the training set? How and why was the tiny images dataset chosen to augment the training set? If we instead used the Gaussian dataset as the OOD training set, will the results have been similar?
2. How large should the OOD training set be? The paper used 100k images from the Tiny Images dataset for the vision tasks. Why 100k? What if we trained with 10k images instead? What should the balance between in-distribution and out-of-distribution images be in the training set?
3. Does the task performance remain the same? If not, then how does the OOD detector (task + OOD classifier) proposed in the paper compare against, say, a model that just learns to classify in-vs-out of distribution?

I look forward to hearing back from the authors and I’m open to changing my score.

---

> ### Author Response · Authors · 2020-11-24
> **Response to AnonReviewer 2**
>
> Thank you for your comments, and we are pleased to hear that you found the paper interesting. Responses below.
>
> > What are the desired characteristics of the OOD dataset used to augment the training set? How and why was the tiny images dataset chosen to augment the training set? If we instead used the Gaussian dataset as the OOD training set, will the results have been similar?
>
> We have experimented with Gaussian, uniform and other un-natural distributions for OoD training data, and the results were poor. Generally, one always gets better results if the exposure is to images that are different -- but not too different -- from the in-distribution. This phenomenon has been discussed in works such as [1]  (which we have detailed comparisons with) and  [2].  The former  — like our method — uses natural OoD images to train, while the latter uses a GAN to generate synthetic images that are plausible but different from the in-distribution. Thus [2] is much more computationally involved than our work.
>
> Using the TinyImage dataset made our method directly comparable to reported results from many competing methods that also used the same protocol. TinyImages is a natural choice when testing with CIFAR-10 and 100 as these datasets (50K each) are actually sampled and curated from the TinyImages set; the latter  consists of ~80 million images and also has many classes not present in the CIFAR sets.
>
>
> > How large should the OOD training set be? The paper used 100k images from the Tiny Images dataset for the vision tasks. Why 100k? What if we trained with 10k images instead? What should the balance between in-distribution and out-of-distribution images be in the training set?
>
> 100K represents a tiny fraction of the 80 million  available samples to choose from, and was a reasonable choice in terms of computational resources, so in our work the ratio between inliers and outliers is 1:2.  Generally, the more outliers the model sees, the better the discriminative power, albeit with quickly diminishing returns. In our experiments, we did not see a noticeable difference between 50K and 100K outlier images, though there was a performance degradation at 10K images.
>
>  While training, in each batch, we keep the number of samples from the outliers to be the same as each inlier class, so the loss on each class (including K+1) is equally weighted.
>
> > Does the task performance remain the same? If not, then how does the OOD detector (task + OOD classifier) proposed in the paper compare against, say, a model that just learns to classify in-vs-out of distribution?
>
> There is a very small, but statistically significant performance drop in classification performance over the actual classes. This might be due to the fact that  our OoD training set  is uncurated, and there are likely images that would also be considered in-distribution. We did not remove these samples (i.e., curate)  for two reasons: one, this is not practical in real-world scenarios, and two, if we did this, we would no longer be able to claim that our method was simple.
>
>
> References:
> 1. Hendrycks et al. Deep Anomaly Detection with Outlier Exposure, ICLR '19
> 2. Neal et al. Open Set Learning with Counterfactual Images, ECCV '18

---

### Decision · Program_Chairs · 2021-01-07
**Final Decision**

**Decision:**

Reject

**Comment:**

This paper proposes a method for out-of-distribution (OOD) detection by introducing a K+1 abstention class for outliers, in addition to the in-distribution classes. While the method has shown promising performance compared to the Outlier Exposure (OE), the novelty is limited given the idea is almost identical to an AAAI'20 paper (Mohseni et al. 2020). In addition, several reviewers have raised concerns regarding the lack of fairness in the experimental setting. Authors are encouraged to address them in a future submission.

The AC believes an interesting and valuable contribution to the community would be showing conceptual and theoretical reasoning for the pros and cons of using K+1 class vs. entropy regularization. Currently, the tradeoff between these two types of training objectives is not well understood.

Overall, three knowledgeable reviewers in this area have indicated rejections. The AC discounted R2's rating due to the less familiarity in this area and lack of participation in the post-rebuttal discussion among reviewers.

Lastly, the AC would like to greatly thank R1, R3, and R4 for the active engagement and participation in the paper discussion period. It was very helpful for the decision-making process.